# Chloroplast genomes of Rubiaceae: Comparative genomics and molecular phylogeny in subfamily Ixoroideae

Serigne Ndiawar Ly[1], Andrea Garavito[2], Petra De Block[3], Pieter Asselman[3,4], Christophe Guyeux[5], Jean-Claude Charr[5], Steven Janssens[3], Arnaud Mouly[6,7], Perla Hamon[1], Romain Guyot[1,8]*

1 Institut de Recherche pour le Développement, UMR DIADE, Université de Montpellier, Montpellier, France, 2 Departamento Ciencias Biológicas, Universidad de Caldas, Manizales, Colombia, 3 Meise Botanic Garden, Meise, Belgium, 4 University of Ghent, Ghent, Belgium, 5 Femto-ST Institute, UMR 6174 CNRS, Université de Bourgogne Franche-Comté, Besançon, France, 6 Laboratory Chrono-Environment, UMR CNRS 6249, Université de Bourgogne Franche-Comté, Besançon, France, 7 Besançon Botanic Garden, Université de Bourgogne Franche-Comté, Besançon, France, 8 Department of Electronics and Automatization, Universidad Autónoma de Manizales, Manizales, Colombia

* romain.guyot@ird.fr

**Data Availability Statement:** All the files are available from the Genbank database (accession

## Abstract

In Rubiaceae phylogenetics, the number of markers often proved a limitation with authors failing to provide well-supported trees at tribal and generic levels. A robust phylogeny is a prerequisite to study the evolutionary patterns of traits at different taxonomic levels. Advances in next-generation sequencing technologies have revolutionized biology by providing, at reduced cost, huge amounts of data for an increased number of species. Due to their highly conserved structure, generally recombination-free, and mostly uniparental inheritance, chloroplast DNA sequences have long been used as choice markers for plant phylogeny reconstruction. The main objectives of this study are: 1) to gain insight in chloroplast genome evolution in the Rubiaceae (Ixoroideae) through efficient methodology for *de novo* assembly of plastid genomes; and, 2) to test the efficiency of mining SNPs in the nuclear genome of Ixoroideae based on the use of a coffee reference genome to produce well-supported nuclear trees. We assembled whole chloroplast genome sequences for 27 species of the Rubiaceae subfamily Ixoroideae using next-generation sequences. Analysis of the plastid genome structure reveals a relatively good conservation of gene content and order. Generally, low variation was observed between taxa in the boundary regions with the exception of the inverted repeat at both the large and short single copy junctions for some taxa. An average of 79% of the SNP determined in the Coffea genus are transferable to Ixoroideae, with variation ranging from 35% to 96%. In general, the plastid and the nuclear genome phylogenies are congruent with each other. They are well-resolved with well-supported branches. Generally, the tribes form well-identified clades but the tribe Sherbournieae is shown to be polyphyletic. The results are discussed relative to the methodology used and the chloroplast genome features in Rubiaceae and compared to previous Rubiaceae phylogenies.

numbers: MN851267-MN851274 and MK577905-
MK577918).

**Funding:** The author(s) received no specific
funding for this work.

**Competing interests:** The authors have declared
that no competing interests exist.

## Introduction

Rubiaceae (coffee family) belongs to Gentianales in the eudicots. It is the fourth most species-rich and diverse family in the flowering plants [1, 2, https://stateoftheworldsplants.org/2017/], comprising ca. 13,600 species grouped in ca. 620 genera and ca. 60 tribes [2, 3]. Rubiaceae are mainly tropical trees and shrubs, and less often annual or perennial herbs [4]. They occupy a large range of ecological niches from desert to evergreen humid forests and from sea level to high altitudes (above 4,000 m [5]). While some herbaceous species reached the temperate regions, Rubiaceae are especially abundant (species diversity and biomass) in lowland humid tropical forest, where they often are the most species-abundant of the woody plant families [2]. The Rubiaceae are divided into two subfamilies, Rubioideae and Cinchonoideae by [1], whereas Bremer and Eriksson [6] recognized three subfamilies, splitting the Cinchonoideae into Ixoroideae and Cinchonoideae.

The pantropical Ixoroideae subfamily comprises ca. 4,000 species [7], distributed into 27 tribes [7, 8, 9], and several well-known genera, i.e. the economically important *Coffea* and the horticulturally important *Gardenia* and *Ixora* [10], besides other less economically important genera such as Vangueria, Alibertia and Duroia L.f.

Molecular phylogenetic analyses of Rubiaceae have been carried out using either nuclear sequences (ETS, ITS, 5S-NTS, pep-C large, pep-V small, PI, Tpi), plastid DNA sequences (*accD-psa1*, *atpB-rbcL*, *ndhF*, *matK*, *petD*, *rbcL*, *rpl16*, *rpl32-trnL*, *rps16*, trnG, *trnH-psbA*, *trnL-F*, *trnT-L*, *trnS-G*) or a combination of both [7, 11, 12, 13, 14]. Altogether, more than twenty markers (fourteen from cpDNA and seven nuclear) have been used for Rubiaceae phylogeny reconstruction, the most popular being ITS and *rbcL*. However, the actual number of amplicons used in individual studies is much lower, e.g. for the dating of the family, subfamily and tribes based on fossils, only five plastid sequences were used [6]. The number of markers used often proved a limitation at tribal and generic levels as authors failed to provide well-supported trees [15, 16]. For instance, Maurin and coworkers [15], using four plastid regions and the internal transcribed spacer (ITS) region of nuclear rDNA (ITS 1/5.8S/ITS 2), failed to get a robust molecular tree for *Coffea*. Similarly, using one plastid and one nuclear marker, Khan and coworkers [17] re-circumscribed the Sabiceeae tribe but were unable to perform a proper biogeographic analysis, given that the molecular tree was largely unresolved. The availability of a robust phylogeny is a prerequisite to accurately study trait evolution at different taxonomic levels (family, subfamily, tribe or genus). This is the case, for instance, when mapping morphological and functional traits in Gardenieae [9], when investigating the evolution of sexual systems and growth habit in Mussaenda [18], or when studying the evolution of caffeine content in *Coffea* [19]. Since two decades, advances in next-generation sequencing (NGS) technologies have revolutionized the field of biology by providing, at reduced cost, huge amounts of data for an increased number of plant species. Among them, short-read sequencing technologies occupy an important place as they need relatively small amounts of DNA (from 600 ng to 1 μg), which allows the use of limited quantities of initial material, such as from herbarium samples [20]. Sequencing on total DNA permits to reconstruct whole chloroplast (cp) genome sequences of around 150–170 kb [21, 22, 23, 24], which can be used to construct robust phylogenies.

Chloroplasts are derived from endosymbiosis between independent living cyanobacteria and a non-photosynthetic eukaryotic host [25, 26]. Most flowering plants, including Coffea species [24] and *Emmenopterys henryi* [23, 27, 28] have a quadripartite circular chloroplast structure with two copies of Inverted Repeat (IR) regions (further called IRA and IRB) separating two regions of unique DNA sequence named large single copy (LSC) and small single copy (SSC) according to their length and gene composition [21]. The comparison of the structure

and gene composition in cp genomes in broad sets of organisms permits to better understand their origin and function [29]. Due to their highly conserved structure, generally recombination-free, and mostly uniparental inheritance, cp DNA sequences have long been used as choice markers for plant phylogeny [30, 31, 32]. However, the low degree of polymorphism among the regular DNA markers used for Rubiaceae phylogenetics often does not resolve relationships at genus level in case of recent speciation [10]. In such conditions, the use of whole cp genome sequences could be a good alternative. In Rubiaceae, complete cp genomes are available for three species of two tribes in subfamily Cinchonoideae (tribe Naucleeae: *Mitragyna speciosa* Korth. [33], *Neolamarckia cadamba* (Roxb.) Bosser [34]; tribe Guettardeae: *Antirhea chinensis* (Champ. ex Benth.) Benth. & Hook.f. ex F.B.Forbes & Hemsl. [35]), five species belonging to at least three tribes in subfamily Rubioideae (tribe Spermacoceae: *Hedyotis ovata* Thunb. ex Maxim. [36]; insertae sedis: *Paralasianthus hainanensis* (Merr.) H.Zhu (as *Saprosma merrillii* H.S.Lo; [37]; tribe Rubieae: *Galium mollugo* L. (NC_028009); tribe Morindeae: *Gynochthodes officinalis* (F.C. How) Razafim. & B.Bremer (as *Morinda officinalis* F.C.How; NC_028009), *Gynochthodes nanlingensis* (Y.Z.Ruan) Razafim. & B.Bremer (NC_028614)], and two species belonging to subfamily Ixoroideae (tribe Condamineae: *Emmenopterys henryi* [23] and tribe Scyphiphoreae: Scyphiphora hydrophyllaceae [38]). However, large projects aiming to develop a library of plastid genomes (including Rubiaceae) are ongoing [33 (GenomeTrakrCP project), 39]).

Nuclear genomic raw data can be assembled into short contigs and used to mine Single Nucleotide Polymorphisms (SNPs) to study the genetic diversity within and between populations and species [40], the evolution of traits of interest [19] or the dynamics of transposable elements [41]. Methodologies based upon short read sequencing such as Genotyping-By-Sequencing (GBS) using a reference genome, permit to define sets of nuclear SNPs for high numbers of genotypes (convenient for multiples of 96 well-plates) as was done for Coffea species [19]. This is possible even with low nuclear genome coverage sequencing (about 10 x coverage). The combination of independent whole genome short read sequencing and bioinformatics tools permit to search these SNPs in different sets of species.

The main objectives of this study were i)- to develop efficient methodology to obtain complete *de novo* assembled cp genomes permitting comparative genomics and a robust molecular phylogenetic tree, ii)- to test the efficiency of mining SNPs in the nuclear genome of non-coffee Rubiaceae based on the use of a coffee reference genome in order to produce a well-supported nuclear tree, and, iii)- to gain insight in chloroplast genome evolution in the Rubiaceae.

## Material and methods

### Material

For this study, we have limited the sampling to subfamily Ixoroideae, to which also *Coffea* belongs. We included 27 taxa representing 10 tribes (Coffeeae, Condamineae, Cordiereae, Gardenieae, Ixoreae, Mussaendeae, Octotropideae, Pavetteae, Sherbournieae and Vanguerieae) plus *Emmenopterys henryi* [23], the complete cp sequence of which was retrieved from NCBI. Detailed information on sampling is given in Table 1.

Our analyses resulted in a single sample, Sherbournia, with a phylogenetic position different from what was expected. In order to verify the identity of this sample, *TrnL-F* and *rps16* sequences were blasted in GenBank. Blasting was then repeated with sequences from other Sherbournia samples obtained with Sanger sequencing. Samples used were *S. bignoniiflora* (Welw.) Hua [Boyekoli Ebale 283 (BR)], *S. buccularia* [Lachenaud et al. 730 (BR)] and *S. zenkeri* Hua [Dessein et al. 1428 (BR)].

Authors of genus and species names of the studied taxa are given in Table 1; for other taxa they are given in the text upon first use.

**Table 1. Taxa studied (species name, tribe, geographic origin, voucher).** [1]according to [1]; [2]according to [6].

| Tribe | Genus | Species | Country | Voucher (collector, collector number, herbarium) | Barcode of herbarium voucher or silica collection (*); accession number of living plant (**) or sequence (***) |
|---|---|---|---|---|---|
| Coffeeae | *Belonophora* Hook.f. | *B. coffeoides* Hook.f. | Cameroon | Dessein et al. 2554 (BR) | BR0000005094424 |
| Coffeeae | Coffea L. | *C. arabica* L. | Ethiopia | NA | ET39**, BRC Bassin-Martin, Reunion |
| Coffeeae | Coffea L. | *C. canephora* Pierre | DR Congo | NA | DH200-94**, BRC Bassin-Martin, Reunion |
| Coffeeae | Coffea L. | *C. sessiliflora* Bridson | Tanzania | NA | PA60**, BRC Bassin-Martin, Reunion |
| Coffeeae | Empogona Hook.f. | *E. congesta* (Oliv.) Hiern | Zambia | Dessein et al. 1103 (BR) | BR6202001591004* |
| Coffeeae | Psilanthus Hook.f. | *P. ebracteolatus* Hiern | Ivory Coast | NA | PSI11**, BRC Bassin-Martin, Reunion |
| Coffeeae | Tricalysia A.Rich. ex DC. | *T. hensii* De Wild. | DR Congo | Boyekoli Ebale 708 (BR) | BR00000012568055 |
| Coffeeae | Tricalysia A.Rich. ex DC. | *T. lasiodelphys* (K.Schum. & K.Krause) A.Chev. | Cameroon | Dessein & Sonké 1462 (BR) | BR0000009955950 |
| Coffeeae | Tricalysia A.Rich. ex DC. | *T. semidecidua* Bridson | Zambia | Dessein et al. 1093 (BR) | BR6202001590007* |
| Coffeeae[1] Bertiereae[2] | Bertiera Aubl. | *B. breviflora* Hiern | Gabon | Champluvier 6182 (BR) | BR0000009043350 |
| Coffeeae[1] Bertiereae[2] | Bertiera Aubl. | *B. iturensis* K.Krause | Gabon | Champluvier 6118 (BR) | BR0000009043206 |
| Coffeeae[1] Bertiereae[2] | Bertiera Aubl. | *B. laxa* Benth. | Cameroon | Dessein et al. 2754 (BR) | BR0000005335817 |
| Condamineae | Emmenopterys Oliv. | *E. henryi* Oliv. | Asia | NA | NC 036300.1*** |
| Condamineae | Pentagonia Benth. | *P. tinajita* Seem. | Costa Rica | Van Caekenberghe 252 (BR) | BR0000009807754 |
| Cordiereae | Alibertia A.Rich. ex DC. | *A. edulis* (Rich.) A.Rich. | Brazil | Van Caekenberghe 485 (BR) | 20121070–69** |
| Gardenieae | Atractocarpus Schltr. & K.Krause | *A. fitzalanii* (F.Muell.) Puttock | Australia | Van Caekenberghe 330 (BR) | BR0000005036035 |
| Gardenieae | Euclinia Salisb. | *E. longiflora* Salisb. | Africa | Van Caekenberghe 348 (BR) | BR0000005036790 |
| Gardenieae | Gardenia J.Ellis | *G.* sp. | Africa | Van Caekenberghe 509 (BR) | 20121077–76** |
| Gardenieae | Schumanniophyton Harms | *S. magnificum* (K. Schum.) Harms | Africa | Van Caekenberghe 499 (BR) | 20090453–07** |
| Gardenieae | Sherbournia G.Don | *S. buccularia* N.Hallé | Cameroon | Lachenaud et al. 736 (BR) | BR0000005336715 |
| Ixoreae | Ixora L. | *I. chinensis* Lam. | Asia | Van Caekenberghe 316 (BR) | BR00009959309 |
| Mussaendeae | Mussaenda Burm. ex L. | *M. pubescens* Dryand. | Asia | Van Caekenberghe 450 (BR) | 20111010–00** |
| Mussaendeae | Pseudomussaenda Wernham | *P. stenocarpa* (Hiern) E. M.A.Petit | DR Congo | Van Caekenberghe 500 (BR) | 20100295–52** |
| Octotropideae | Feretia Delile | *F. aeruginescens* Stapf | Zambia | Dessein et al. 912 (BR) | BR0000009819672 |
| Pavetteae | Leptactina Hook.f. | *L. leopoldi-secundi* Büttner | Congo | Champluvier 5428 (BR) | BR0000008566447 |
| Pavetteae | Pavetta L. | *P. schumanniana* F. Hoffm. ex K.Schum. | DR Congo | Malaisse 13702 (BR) | BR0000006430252 |
| Pavetteae | Tarenna Gaertn. | *T. grevei* (Drake) Homolle | Madagascar | De Block et al. 959 (BR) | BR0000009125964 |
| Sherbournieae | Mitriostigma Hochst. | *M. axillare* Hochst. | Africa | Van Caekenberghe 44 (BR) | BR0000006429812 |
| Vanguerieae | Vangueria Juss. | *V. infausta* Burch. | Zambia | Dessein et al. 879 (BR) | BR6202005552001* |

## Genome sequencing

Whole genomic DNA was isolated from silica or living plant material following a modified cetyltrimethylammonium bromide (CTAB) method [42]. A total of 25 mg (silica dried) or 100 mg (fresh) leaf material was ground into a fine powder. To eliminate secondary metabolites, two consecutive chloroform cleaning steps were carried out. DNA was lysed either in elution buffer (10 mM Tris-HCl, pH 8.0–8.5) or sterile PCR-grade water. The use of EDTA in elution buffer should be avoided to circumvent possible enzymatic inhibition in downstream applications which may lead to lower library quality. The short-read sequencing was done using the BGI-seq 500 platform, 2x100 bp paired-end. The quality of reads was verified using the Java software FASTQC (https://www.bioinformatics.babraham.ac.uk/projects/fastqc/). The raw data were checked in order to detect potential contamination using Kraken and Krona tools [43]. Raw reads were cleaned when necessary using Trimmomatic [44].

## Chloroplast genome assembly and annotation

Cp genomes were *de novo* assembled using NOVOplasty software [45] from the sorted cp raw sequences obtained. Good quality raw reads were split into two sets corresponding to forward (F) and reverse (R) reads. With the aim to sort only cp data, the two sets of data (F and R) were mapped against the *Coffea arabica* cp reference genome [46] using Bowtie2 [47]. The choice of *C. arabica* (the genetically best-known species among Rubiaceae) at this step is justified since one objective of this study is to test the transferability of tools and methodology developed for *Coffea* to other members of Rubiaceae. Then, *in silico* filtered reads were considered for the cp genome *de novo* assembly using NOVOplasty. The recalcitrant cp genomes were re-assembled using Abyss [48]. At the end of the assembly process, the cp genomes were compared with the reference *C. arabica* genome using Gepard [49] in order to check for incongruences of the assembly. To confirm the overall structure, the pair-end illumina reads are mapped back to the assembly using Bowtie2 [47] and BAM files are used to display the read coverage using Artemis (https://www.sanger.ac.uk/science/tools/artemis-comparison-tool-act). The cp genomes were then annotated using Geseq [50] as recommended by Guyuex et al., [24]. The circular visualization of each cp genome was obtained using the OrganellarGenomeDRAW tool (OGDRAW) [51]. The linear gene order comparison was obtained using ACT (Artemis Comparison Tool [52]).

## Sequence divergence and junction sequences divergence

Given that *Coffea arabica* could not be used as outgroup in the phylogeny, we decided to use the annotated genomes of *Antirhea chinensis* [35], *Mitragyna speciosa* [33] and *Neolamarckia cadamba* [34] belonging to the Cinchonoideae subfamily as reference genomes and outgroup taxa for the cp genome analyses. The alignments of the complete chloroplast genome sequences of the 28 studied Rubiaceae were visualized using mVISTA [53] in order to show global interspecific variation and variation within the tribes.

Taking into account data obtained from for each taxon (length of regions LSC, SSC, IR and gene annotation), we calculated the distance between boundaries and the nearest gene to visualize junction sequence divergence between species and within tribes.

## Phylogenetic relationships

The plastid phylogeny was produced using a total of 28 cp sequences and one of three outgroups belonging to the Cinchonoideae subfamily (*Antirhea chinensis*, *Mitragyna speciosa* and *Neolamarckia cadamba*, retrieved from GenBank). The sequences were first aligned using MAFFT 7.305 with the following parameters BLOSUM62 and 200PAM/ k = 2 [54].

The nuclear tree was produced using a total of 27 taxa (no data available for *Emmenopterys henryi*). No non-Ixoroideae data were available, so the trees were rooted midpoint. The 28,800 SNPs used for *Coffea* [19] were mined according to the methodology developed by these authors. In a second step, in order to reduce the amount of missing data, rare or too common sites were removed using Tassel ver. 5.0 [55] with the following parameters: minimum count = 18, minimum frequency = 0.2, maximum frequency = 0.8. A total of 1,726 sites (SNPs) were kept.

MAFFT alignment of cp sequences and nuclear SNP concatenation were used to infer the phylogenetic trees. The phylogenetic reconstructions were done using RAxML version 8 [56] under the General Time Reversible nucleotide substitution model with gamma distributed rate variation among sites (GTR+G), ML estimate of alpha-parameter, BFGS method to optimize GTR rate parameters and Felsenstein's bootstraps option autoMRE as recommended by the author). The trees were then edited with FigTree ver. 1.3.1 [57] and Inkscape (https://inkscape.org/fr/release/0.91/).

## Results and discussion

### Chloroplast genome features in Ixoroideae

Among the 28 studied samples 25 exhibited the classical quadripartite structure but three had an apparent tripartite structure with only one IR (*Mussaenda pubescens*, *Feretia aeruginescens* and *Pavetta schumanniana*). These latter belong to three different tribes (Mussaendeae, Octotropideae and Pavetteae, respectively) but the tripartite structure is not present in all representatives of these tribes.

Regarding the quadripartite genomes, total length ranges from 153,056 bp for *Bertiera breviflora* to 155,328 bp for *Sherbournia buccularia*. Similar length differences are observed among the tripartite genomes (from 127,396 bp for *Pavetta schumanniana* to 129,508 bp for *Mussaenda pubescens*). Length variations were also noted for the different regions: from 83,406 bp (*Vangueria infausta*) to 85,461 bp (*Belonophora coffeoides*) for LSC, from 17,915 bp (*Mitriostigma axillare*) to 18,245 bp (*Emmenopterys henryi*) for SSC and from 24,855 bp (*Bertiera laxa*) to 25,978 bp (*Mussaenda pubescens*) for IR. In all species, GC content was similar for the complete cp genome (ca. 37%) as well as for each of the cp subregions (LSC: ca. 35%; SSC: ca. 31%; IR: ca. 43%). Individual information is summarized in Table 2.

We annotated a total of 118 different genes belonging to 14 functional categories and present in all the genomes with the exception of *Tarenna grevei* which has lost *trnH-GUG*. One hundred genes were present as single copy, 17 were duplicated and one (*rps12*) was triplicated in IR. The LSC, SSC and IR regions contained 87, 13 and 18 genes respectively. Among the 80 protein-coding genes identified, only nine include introns. Seven of these contain one intron (*atpF*, *ndhA & B*, *rpoC1*, *rps12*, *rps16*, *rpl2*) whereas *clpP* and *ycf3* have two introns. Complete *infA* and *pbf1* genes were present in all studied species. For the three taxa showing only one IR, the corresponding genes were present in only one copy.

The annotated cp genome sequences permitted to compare the length of the junctions of the main regions LSC, IR and SSC among the studied Rubiaceae (Table 3). Generally, low variation was observed between the taxa in the boundary regions. However, while the distance for LSC/IRB junctions was 88 bp for most species, variation was noted in *Emmenopterys henryi* (Condamineae) with 30 bp, in *Psilanthus ebracteolatus* (Coffeeae) with 358 bp, in *Coffea sessiliflora* (Coffeeae) with 157 bp and in *Sherbournia buccularia* (Gardenieae) with 148 bp. A similar variation was obtained for the IRB/SSC junction. (S1 Table)

Data obtained for other Rubiaceae such as *Hedyotis ovata* [36] and *Paralasianthus hainanensis* (as *Saprosma merrillii*; [37]) from the Rubioideae subfamily and *Antirhea chinensis* [35]

**Table 2. Chloroplast genome main features of the 28 studied taxa ordered alphabetically within 10 tribes.**

| Species name | number of IR | Length in bp | | | |
|---|---|---|---|---|---|
| | | Genome | LSC | SSC | IR |
| **Coffeeae** | | | | | |
| *Belonophora coffeoides* | 2 | 155190 | 85461 | 18135 | 25797 |
| *Coffea arabica* | 2 | 155186 | 85157 | 18139 | 25945 |
| *Coffea canephora* | 2 | 154982 | 85109 | 21297 | 24288 |
| *Coffea sessiliflora* | 2 | 155010 | 85100 | 18110 | 25900 |
| *Empogona congesta* | 2 | 154672 | 85106 | 18182 | 25692 |
| *Psilanthus ebracteolatus* | 2 | 155084 | 85134 | 18142 | 25904 |
| *Tricalysia hensii* | 2 | 154953 | 85407 | 18166 | 25690 |
| *Tricalysia lasiodelphys* | 2 | 154898 | 85418 | 18138 | 25665 |
| *Tricalysia semidecidua* | 2 | 154816 | 85338 | 18166 | 25656 |
| **Coffeeae/Bertiereae** | | | | | |
| *Bertiera breviflora* | 2 | 153055 | 85231 | 21974 | 22925 |
| *Bertiera iturensis* | 2 | 154675 | 85399 | 18172 | 25552 |
| *Bertiera laxa* | 2 | 153778 | 85469 | 17981 | 25164 |
| **Condamineae** | | | | | |
| *Emmenopterys henryi* | 2 | 155379 | 85554 | 18245 | 25790 |
| *Pentagonia tinajita* | 2 | 153604 | 84822 | 18106 | 25338 |
| **Cordiereae** | | | | | |
| *Alibertia edulis* | 2 | 154508 | 84692 | 18138 | 25839 |
| **Gardenieae** | | | | | |
| *Atractocarpus fitzalanii* | 2 | 154627 | 84991 | 17930 | 25853 |
| *Euclinia longiflora* | 2 | 155182 | 85363 | 18181 | 25819 |
| *Gardenia sp.* | 2 | 155294 | 85475 | 18127 | 25846 |
| *Schumanniophyton magnificum* | 2 | 155081 | 85386 | 18115 | 25790 |
| *Sherbournia buccularia* | 2 | 155328 | 85529 | 18171 | 25814 |
| **Ixoreae** | | | | | |
| *Ixora chinensis* | 2 | 154665 | 84874 | 18157 | 25817 |
| **Mussaendeae** | | | | | |
| *Mussaenda pubescens* | 1 | 129508 | 85411 | 18118 | 25979 |
| *Pseudomussaenda stenocarpa* | 2 | 155057 | 85189 | 18018 | 25925 |
| **Octotropideae** | | | | | |
| *Feretia aeruginescens* | 1 | 129434 | 85285 | 18212 | 25937 |
| **Pavetteae** | | | | | |
| *Leptactina leopoldi-secundi* | 2 | 154462 | 84936 | 18222 | 25652 |
| *Pavetta schumanniana* | 1 | 127401 | 83569 | 18033 | 25796 |
| *Tarenna grevei* | 2 | 154164 | 84420 | 18124 | 25810 |
| **Sherbournieae** | | | | | |
| *Mitriostigma axillare* | 2 | 153606 | 84967 | 17915 | 25362 |
| **Vanguerieae** | | | | | |
| *Vangueria infausta* | 2 | 152987 | 83406 | 18019 | 25781 |

from the Cinchonoideae subfamily indicate a quadripartite structure and a total of 114 genes (eight duplicated genes counted once and eight genes missing in Rubiaceae, see below) of which 80 are unique protein-coding genes. For *Neolamarckia cadamba* (Cinchonoideae), [34] revealed a total of 130 genes, 79 of which are protein-coding. Data obtained from GenBank for three Rubioideae (accession numbers NC_036970 for *Galium mollugo*, NC_028009 for

**Table 3. Plastid genomes features of three taxa from the Rubioideae and three taxa from the Cinchonoideae subfamilies.** Estimations were done from data extracted from [33, 34, 35, 59] or calculated from data extracted from GenBank for *Morinda officinalis* (NC_028009) and *Gynochtodes nanlingensis* (NC_028614).

| Species name | number of IR | Length in bp | | | | GC content (%) | | | |
|---|---|---|---|---|---|---|---|---|---|
| | | total genome | LSC | SSC | IR | Overall | LSC | SSC | IR |
| **Rubioideae subfamily** | | | | | | | | | |
| *Morinda officinalis* | 2 | 153398 | 84011 | 17855 | 25766 | 36 | 35 | 31 | 43 |
| *Gynochtodes nanlingensis* | 2 | 154086 | 84329 | 18115 | 25821 | 37 | 35 | 31 | 43 |
| *Galium mollugo* | 2 | 153677 | 84471 | 17056 | 26075 | 37 | 35 | 31 | 43 |
| **Cinchonoideae subfamily** | | | | | | | | | |
| *Antirhea chinensis* | 2 | 155616 | 86252 | 17984 | 25690 | 38 | 36 | 31 | 43 |
| *Mitragyna speciosa* | 2 | 155600 | 86213 | 18201 | 25593 | 37 | 35 | 32 | 43 |
| *Neolamarckia cadamba* | 2 | 154,999 | 85880 | 17851 | 25634 | 38 | 35 | 32 | 43 |

*Morinda officinalis* and NC_028614 for *Gynochtodes nanlenginsis*) and from literature for two Cinchonoideae [34, 35] permitted us to determine their plastid features (Table 3). All have the classical quadripartite structure. In the Rubioideae, the total cp length varies from 153,398 to 154,086 bp; LSC from 84,011 to 84,471 bp; SSC from 17,056 to 18,115 bp and IR from 25,766 to 26,075 bp. In the Cinchonoideae, total cp length varies from 154,999 to 155,616 bp; LSC from 85,880 to 86,252 bp; SSC from 17,851 to 17,984 bp and IR from 25,634 to 25,690 bp. The GC content for the total sequence and for the different regions are similar to our results in Ixoroideae. The chloroplast genome features of the Ixoroideae, and of the Rubiaceae as a whole, are in the ranges reported for most flowering plants [29, 30, 58].

The tripartite genome structure was not yet reported in the Rubiaceae but was recorded for Fabaceae [60], Geraniaceae [61], Pinaceae [62], Cactaceae [63], Arecaceae [64] and Passiflora-ceae [65]. Within the Ixoroideae, the chloroplast assembly of three species showed a tripartite genome structure. Besides frequent inversions, duplications, or losses of fragments [65, 66], IR expansion/contraction and even IR absence contributed to substantial variation in cp genome length [67]. However, the robustness of our assemblies of species showing only one IR were tested. All reads were mapped on the assembly and the read coverage was displayed. The read coverage showed an increase at the IR region of the assembly suggesting that two IR regions may be present but assembled into only one IR (S1 Fig). So, the assembly of these chloroplast sequences (*Mussaenda pubescens, Feretia aeruginescens* and *Pavetta schumanniana)* and the IR absence should be considered with caution since this event is very rare in most other plant families [68] and since we cannot exclude that the assembly process collapsed the IR regions into only one. Increasing the number of taxa investigated and using long read sequencing techniques, such as PacBio and Oxford Nanopore may demonstrate it to be a not so rare event in the Rubiaceae, which would lead to questions on the role of two or one IR in the evolution of land plants.

Circular visualization of the *Bertiera breviflora* cp genome is given in Fig 1 as an example. Gene order and orientation from pairwise comparisons were generally well-conserved although some gene orientations were different (S2 Fig).

Sequence divergence was visualized using mVISTA with *Coffea arabica* as the reference annotated genome. The choice of *Coffea arabica* instead of *Antirhea chinensis* (outgroup used for the plastid phylogeny) was justified by the level of divergence between Cinchonideae and Ixoroideae. Globally, sequence divergence among all taxa was relatively high and mainly concentrated in conserved non-coding sequences and in Untranslated Transcribed Regions (UTR). However, variation among species seemed to be negligible for UTRs located in the IR region (*rpl2*, *ndhB* and *rps12* genes). Substitutions were more frequent but indels were observed as well, even in the *ycf2* exon. Four (the conserved non-coding regions between

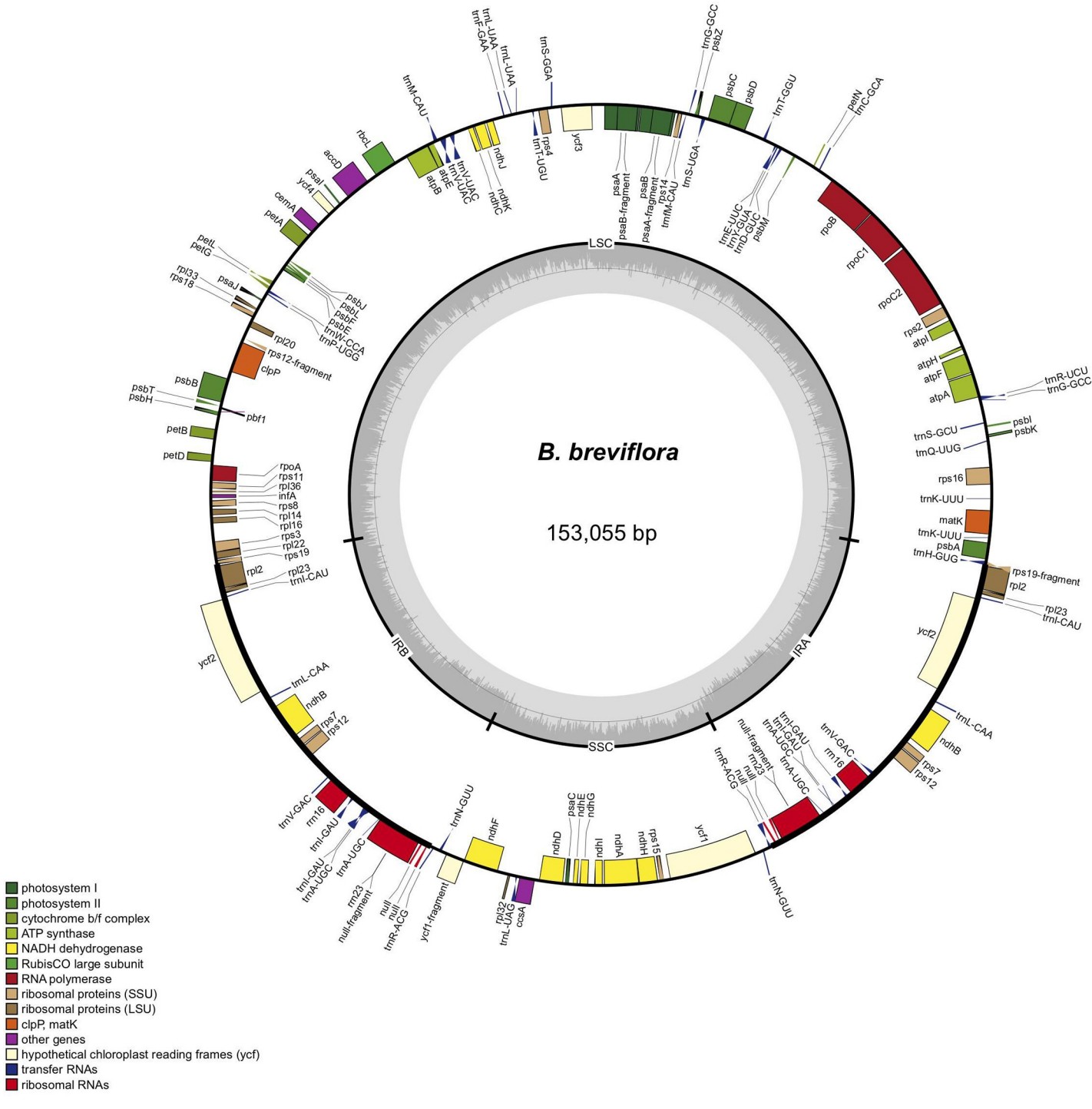

**Fig 1. Circular visualization of annotated Rubiaceae genomes showing the quadripartite structure of *Bertiera breviflora* (similar to 25 taxa studied here).**

*matK* and *atpA*, *rpoB* and *psbD*, *rps4* and *ndhJ* and *ndhC* and *atpE*), and two (*ndhF—ccsA* and *ycf1*) hypervariable regions were identified in the LSC and SSC regions respectively. A representation of sequence divergence is given for a selected set of taxa (Fig 2). In total, 31.5% sites of the complete alignment included indels.

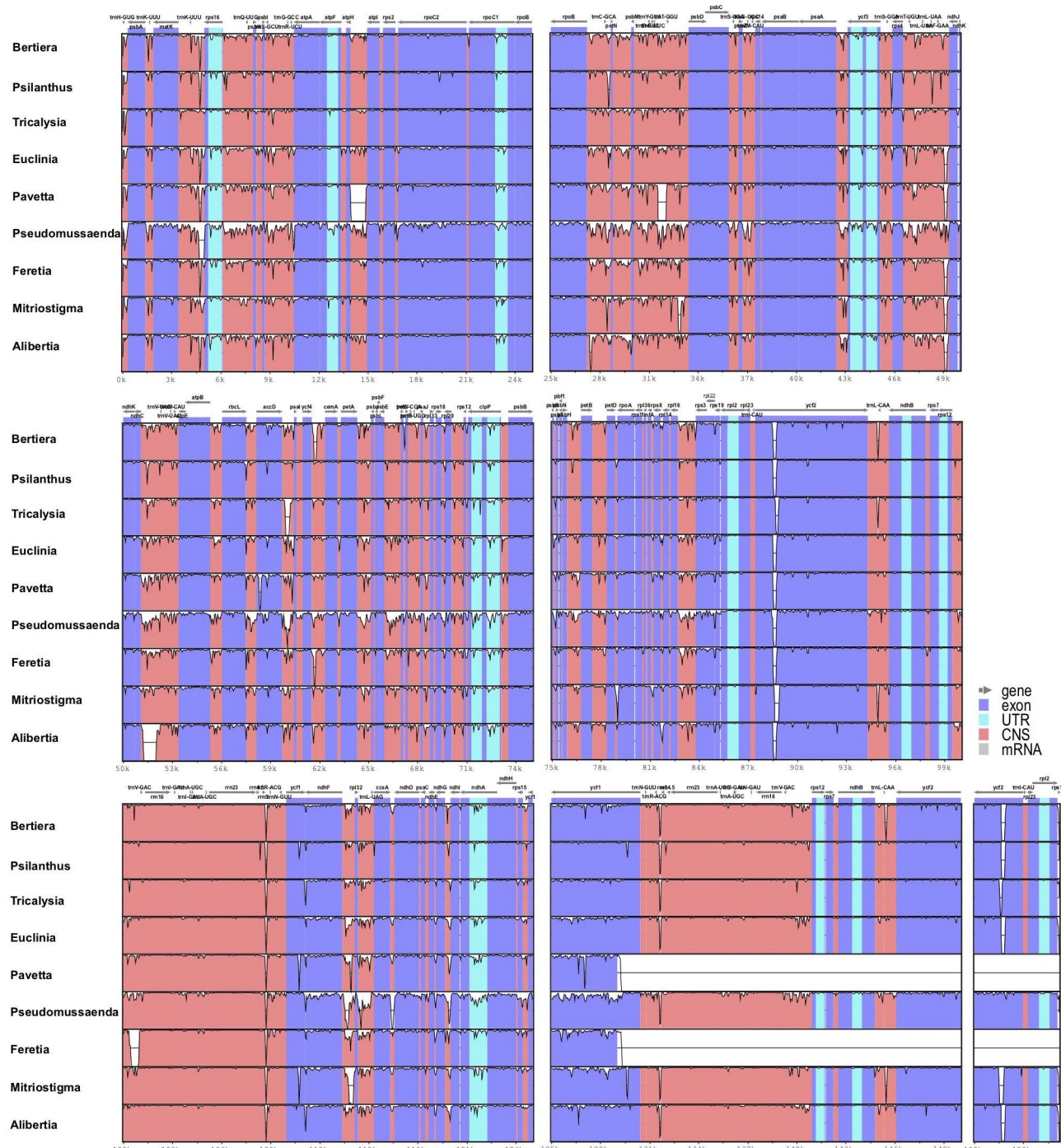

**Fig 2. Sequence identity plot comparing nine species of subfamily Ixoroideae with *Coffea arabica* as the annotated reference genome using mVISTA.** The location and orientation of the genes are indicated on the top. Exons and UTRs are in purple and turquoise respectively. Conserved non-coding regions are in orange. The y-axis ranges from 100% (top) to 50% identity between each sequence and the reference. The order of the taxa used from top to bottom is: *Bertiera iturensis* (Bertiereae), *Psilanthus ebracteolatus* (Coffeeae), *Empogona congesta* (Coffeeae), *Euclinia longiflora* (Gardenieae), *Pavetta schumanniana* (Pavetteae), *Pseudomussaenda stenocarpa* (Mussaendeae), *Feretia aeruginescens* (Octotropideae), *Mitriostigma axillare* (Sherbournieae) and *Alibertia edulis* (Cordiereae). *Feretia aeruginescens* and *Pavetta schumanniana* showing only one IR in the current assembly.

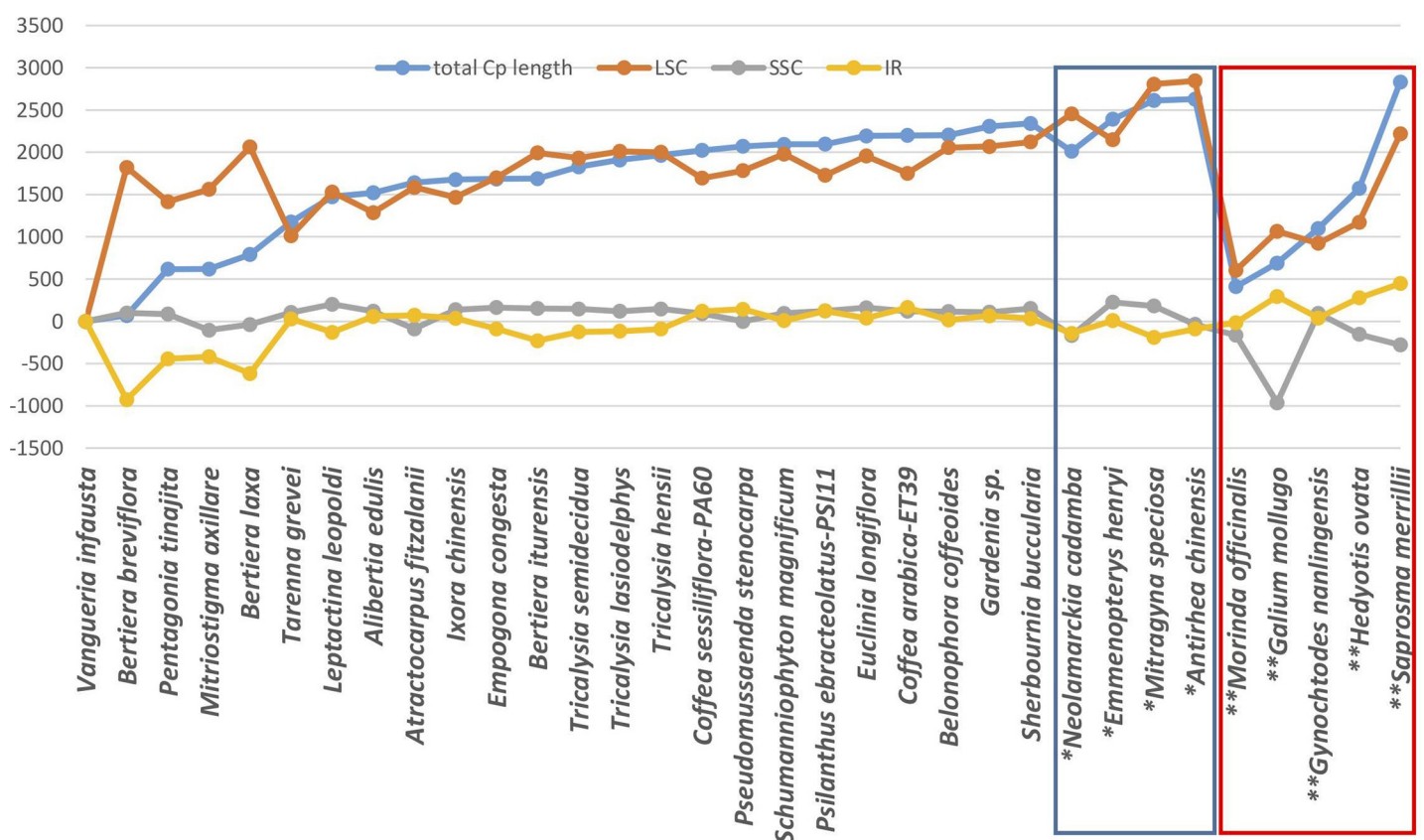

**Fig 3. Variation in the length of the different regions [y-axis values are minus data for the smallest cp genome total length (*Vangueria infausta*)].** The taxa are ordered in increasing total cp genome size in each subfamily (24 Ixoroideae, four Cinchonideae in blue box marked with one asterisk and five Rubioideae in red box marked with two asterisks). Data for taxa indicated with asterisk(s) was retrieved from literature [23, 33, 34, 35, 36, 37] or calculated from data extracted from GenBank for *Morinda officinalis* (NC_028009), *Gallium mollugo* (NC_036970), *Gynochtodes nanlingensis* (NC_028614).

In Ixoroideae, plotting the length variation of the different regions relative to the smallest cp genome (here *Vangueria infausta* and considering only the quadripartite cp genomes*),* showed the pattern of variation given in Fig 3. The length of the different regions did not increase simultaneously to the total cp length except for the smallest four cp genomes. The increase in size seems to be mainly due to increase in length of LSC and possibly to gene and/ or intron length increases. For *Bertiera breviflora* and *Bertiera laxa*, the increase in cp size is mainly due to the increase in length of LSC associated with a decrease in length of IR. Variation in the length of SSC has only limited impact on cp size variation. Regarding the four Cinchonoideae and the five Rubioideae species, similar patterns of variation are observed with the exception of *Galium mollugo* for which a decrease in length of SSC is notable. Therefore, with the exception of a few species, it seems that length variation in LSC is the main contributor to cp size variation. Among eudicots, the progressive expansion of the IR has been documented in Pelargonium L'Hér. ex Aiton [69] and Passiflora L. [65], and a similar molecular mechanism driving the IR evolution in these two unrelated lineages could be a possibility.

Angiosperm cp genomes exhibit a remarkably conserved gene content and order as observed for instance within Fagaceae [21, 70, 71, 72] and more specifically for Quercus L. [22]. Likewise, gene content and order were nearly identical in the Ixoroideae representatives studied as well as in representatives of the two other Rubiaceae subfamilies.

Recorded in tobacco and in most others members of Solanaceae as a pseudogene [46], *infA* was intact in all Ixoroideae species of this study and in the Rubioideae and Cinchonoideae species for which whole cp genomes are available. Similarly, putatively involved in photosystem I and II biogenesis, *pbf1* (*psbN* in *Coffea arabica*, [46]) was present in all Ixoroideae as well as in the Rubioideae and Cinchonoideae. In all Ixoroideae of this study and in the species of the other subfamilies, a fragment of *rps19* appeared duplicated at the IR/LSC boundaries as reported in Solanaceae with the exception of tobacco [73]. Eight genes (*CHLB, CHLL, CHLN, CYSA, CYST, MBPX, PSAM,* and *RPL21)* were absent in the study of 16 wild coffee trees [24]. This study showed their absence in all Ixoroideae and the other Rubioideae and Cinchonoideae tested. Finally, despite minor changes in gene content, orientation and order, Ixoroideae plastid genomes are well conserved within and between tribes. This was also the case in the available Cinchonoideae and Rubioideae species and, therefore, could be true for the whole family. However, sequence divergence within and between tribes was observed and at much higher level (Fig 2) than reported in Quercus [22].

## Plastid molecular phylogeny and comparison to previous Rubiaceae phylogenies

The complete cp genome-based phylogeny included 28 Ixoroideae taxa and *Antirhea chinensis* (Cinchonoideae subfamily) as outgroup. Maximum Likelihood analyses resulted in a generally well-resolved topology with highly supported branches, except for four lineages: the branch between Empogona and the Belonophora/Tricalysia clade, the branch between Leptactina and Pavetta/Tarenna within the *Pavetteae* tribe, the branch towards the Cordiereae/Octotropideae clade and the branch towards the Mussaendeae/Condamineae clade (BS < 80%, Fig 4). The ingroup has three main clades: Mussaendeae (Fig 4; in green), Condamineae (Fig 4; in red) and a large clade comprising all other taxa. The Mussaendeae and Condamineae are well-supported as distinct monophyletic lineages (BS = 100) but their mutual relationship and their relationship with the rest of the ingroup remain unclear. The rest of the ingroup forms a well-supported clade (BS = 100) and comprises two well-supported subclades (BS = 100) that correspond to the Vanguerieae alliance (Ixora and Vangueria) and the Coffeeae alliance. Within the Coffeeae alliance, the tribe Pavetteae (Fig 4; in pink), the Coffeeae/Bertiereae lineage with Bertiera sister to the Coffeeae (Fig 4; in blue) and the clade comprising Schumanniophyton, Gardenia, Sherbournia, Euclina and Atractocarpus (Gardenieae, Fig 4; in brown) are supported as monophyletic groups (BS = 100). All tribes represented by at least two representatives are retrieved as monophyletic with the exception of the Sherbournieae. Sherbournia does not form a clade with Mitriostigma but is firmly embedded in Gardenieae. The same phylogeny was obtained with *Mitragyna speciosa* (Cinchonoideae) as outgroup. However, when using *Neolamarckia cadamba* as outgroup, a slightly different phylogenetic tree was obtained (data not shown).

This chloroplast phylogeny concurs well with previously published phylogenetic trees based on Sanger sequencing of several markers [1, 6, 7]. Within the Ixoroideae (Ixoridinae sensu Robbrecht and Manen), Robbrecht and Manen [1] recognized a basal clade (basal Ixoridinae; not represented in our analysis) and two main evolutionary lineages Ixoridinae I and Ixoridinae II. Ixoridineae I is essentially neotropical and represented here by the tribe Condamineae. Ixoridineae II is mainly paleotropical and includes all other members of the ingroup. Unlike the super-tree of Robbrecht and Manen [1], our plastid phylogeny is not resolved at the base and does not clearly separate Ixoridineae I and II, since Mussaendeae is considered part of Ixoridineae II by [1]. Bremer and Eriksson [6] did not distinguish lineages within the subfamily Ixoroideae, probably because the base of their phylogenetic tree is unresolved. Kainulainen

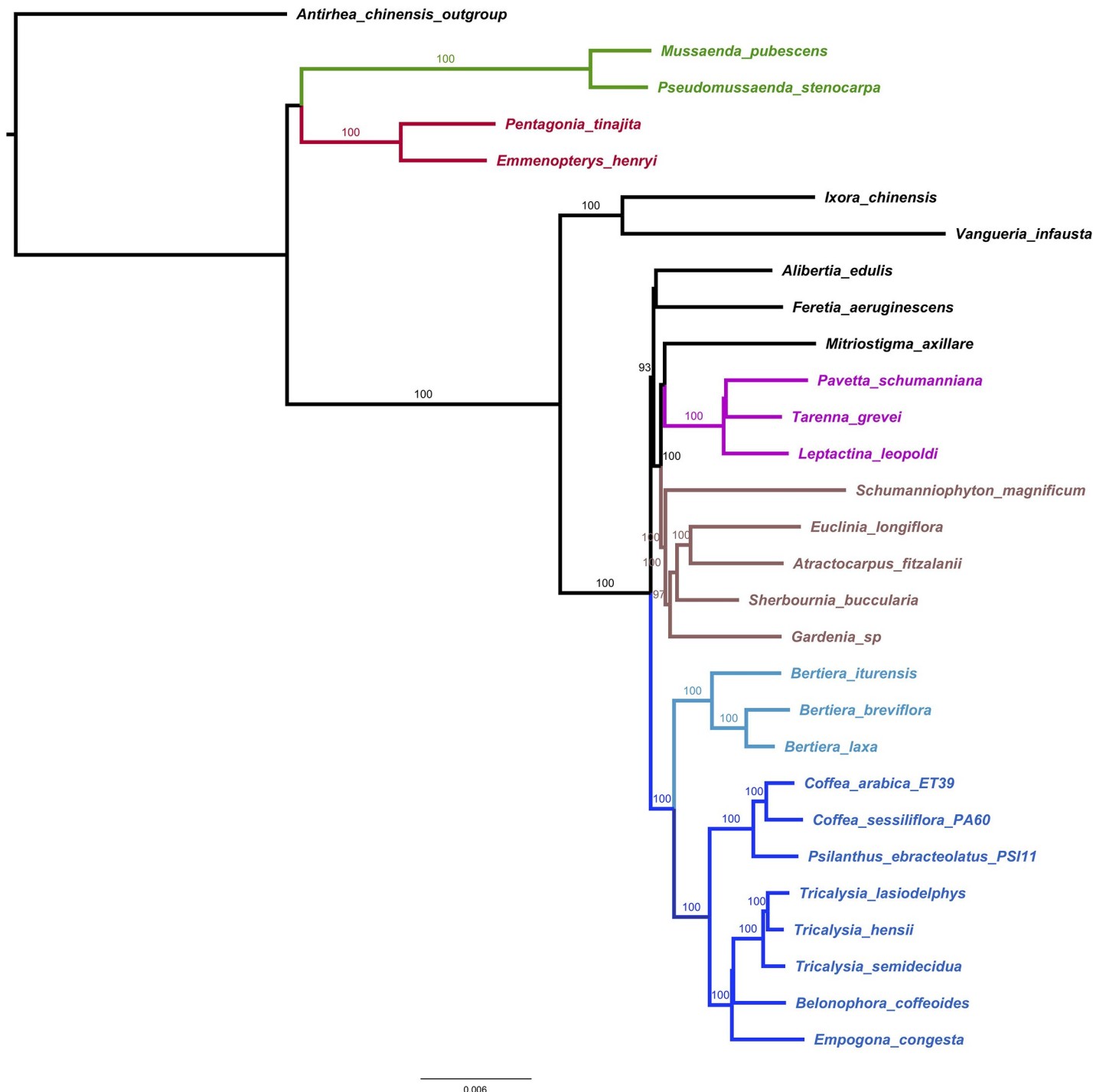

**Fig 4. Maximum likelihood plastid tree (RAxML with GTR model of substitution) based on the whole cp sequences of 28 Ixoroideae (with *Antirhea chinensis* as outgroup) and bootstrap values to estimate the branch support.** Four well-supported clades are marked in green for Mussaendeae, red for Condamineae, pink for Pavetteae, brown for Gardenieae and blue for Coffeeae/Bertiereae.

et al., [7] recognized within the Ixoroideae a basal grade (here represented by Condamineae and Mussaendeae) and a clade of core Ixoroideae. Our analysis confirms the subfamilial classification of Kainulainen et al., [7] rather than that of Robbrecht and Manen [1]. Within the

core Ixoroideae, Kainulainen et al., [7] differentiated between the Vanguerieae alliance and the Coffeeae alliance. These two clades are also retrieved in our analysis. Our analysis confirms the sister relationships between Ixoreae and Vanguerieae [1, 6, 7] and between Coffeeae and Bertiereae [1, 6, 7, 9].

Gardenieae is retrieved as monophyletic only with the inclusion of Sherbournia, which has been considered part of the tribe Sherbournieae [9]. The Sherbournieae were recently instated [9] to include the former Gardenieae genera Sherbournia, Mitriostigma, Atractogyne and Oxyanthus, the last two of which are not included in our analysis. With the exception of Sherbournia, these genera are characterized by pollen grains in tetrads [73]. Persson and more recently Bremer and Eriksson [74, 6] also retrieved this group of three genera with pollen in tetrads as monophyletic. However, the inclusion of Sherbournia makes the tribe morphologically heterogeneous as regards to pollen characters (pollen in monads). In order to check the identity of our Sherbournia sample we separated *TrnL-F* and *rps16* sequences from the whole genomes sequence and blasted them in GenBank, where they showed more similarity with Rothmannia Thunb. than with the Sherbournia sequences present there. This was repeated with sequences from other Sherbournia species obtained with Sanger sequencing with the same results. We are therefore confident that Sherbournia does not form part of the tribe Sherbournieae as delimited by Darwin [9], but belongs to the Gardenieae. It should be noted that also in the phylogeny of Persson [74], Sherbournia groups with Rothmannia. The tribe Gardenieae has been demonstrated in several studies to be polyphyletic [1, 7, 9]. The fact that it is not so in our analysis is the result of the small number of representatives included, notably five genera out of over fifty [9]. The five genera making up the Gardenieae clade in our analysis are not generally considered closely related. Atractocarpus is part of Gardenieae IV in [1] and of the Porterandia group in [9], *Gardenia* is part of Gardenieae II [1] and the *Gardenia* group [9], Euclinia belongs to Gardenieae III [1] and the Randia group [9], Schumanniophyton belongs to Gardenieae I [1] and remains unplaced in [9] and Sherbournia is unplaced in [1] and belongs to the Sherbournieae tribe in [9].

## Nuclear SNP mining and Efficiency of transferability of methods from Coffea to Ixoroideae

The genome of *Coffea canephora* Pierre ex A.Froehner was used as reference genome to mine SNPs as described in Hamon et al. (2017). This methodology was efficient despite unequal results between taxa. No outgroup from another subfamily of the Rubiaceae was available so the tree was rooted midpoint. In this analysis *Coffea canephora* was added but *Emmenopterys henryi* could not be included since no nuclear genome data was available.

An average of 22,906 SNPs was sorted with the extremes ranging from 10,335 in *Pentagonia tinajita* to 27,642 in *Tricalysia lasiodelphys*. Among the 806,400 individual data expected (28 x 28,800), excluding all Coffea *species*, the average percentage of missing data was 31% ranging from 10% in *Tricalysia hensii* to 77% in *Ixora chinensis* and *Pentagonia tinajita*. The percentage of heterozygotes was 0.6% on average but varied from 0.25% in *Atractocarpus fitzalanii* to 3.2% in *Psilanthus ebracteolatus*. The nucleotide percentage was 29.1% for A, 29.3% for T, 20.5% for G and 20.2% for C (S1 Appendix). So, the SNP transferability from Coffea [19] to non-coffee Rubiaceae belonging to ten tribes of subfamily Ixoroideae can be considered as successful. Interestingly, the phylogenetically most distant species are those with the fewest orthologous sequences.

The species relationships obtained with the complete dataset (28,800 sites) are shown in Fig 5. The Maximum Likelihood tree shows a majority of well-supported branches (BS of 86–100%). The ingroup shows two main, well-supported clades, the first comprising the Coffeeae

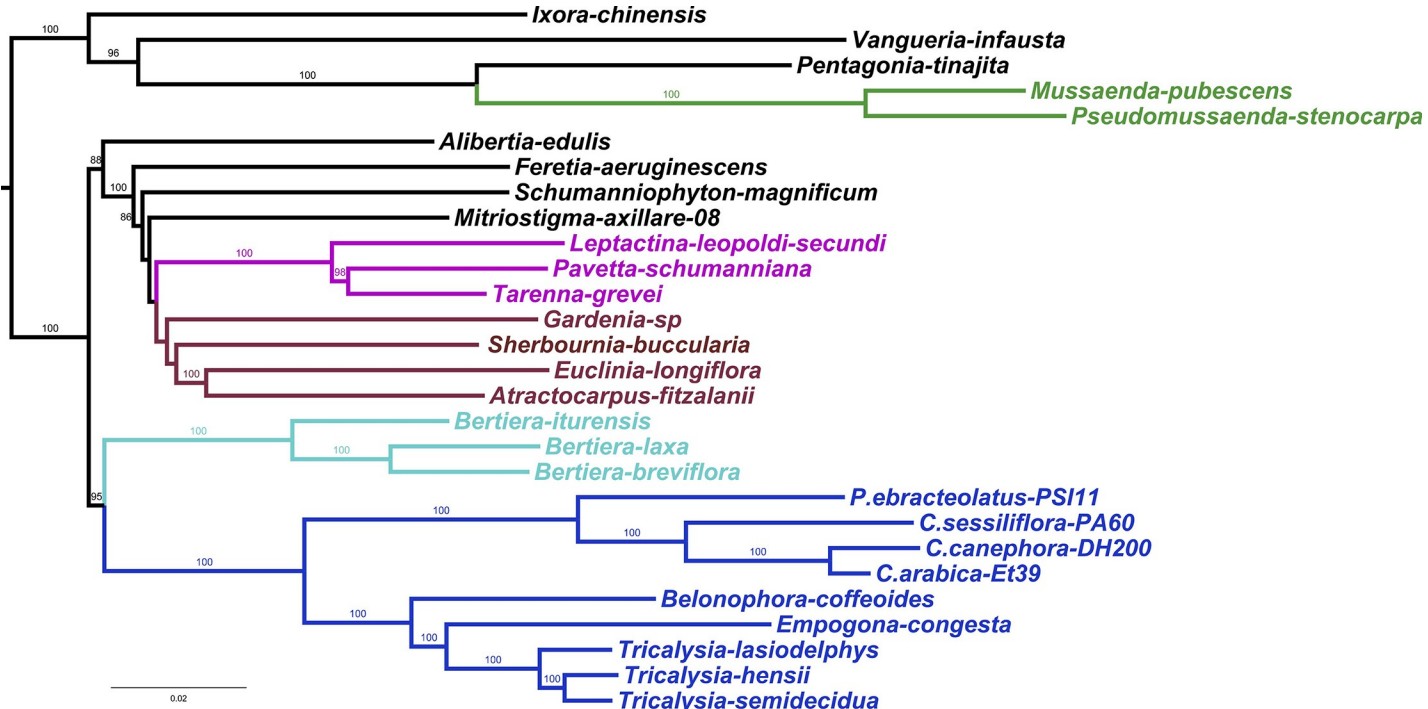

**Fig 5. Maximum likelihood nuclear tree of 28 Ixoroideae based on 28,800 SNPs (RAxML with GTR model of substitution) and bootstrap values to estimate branch support.** Colored clades indicated well-defined tribes. The tree is rooted midpoint as no outgroup is available. Green for Mussaendeae; pink for Pavetteae; brown for Gardenieae; blue for Coffeeae and turquoise for Coffeeae/Bertiereae.

alliance and the second comprising the Vanguerieae alliance, the Condamineae and the Mussaendeae. The Mussaendeae (Fig 5; in green), Pavetteae (Fig 5; in pink), Coffeeae/Bertiereae (Fig 5; in blue) and Gardenieae (Fig 5; in brown) are supported as monophyletic with high branch support values. Bertiera is sister to the Coffeeae. The following results are in contrast to the results of the plastid phylogeny: the Gardenieae clade (Fig 5; in brown) does not include Schumanniophyton; Ixora and Vangueria (Vanguerieae alliance) do not form a monophyletic group. The monophyly of the Condamineae cannot be evaluated because only a single representative is present in this analysis (no data for *Emmenopterys henryi*). The tribe Sherbournieae (Sherbournia and Mitriostigma) is not retrieved as monophyletic and it is embedded in the Gardenieae clade.

The phylogenetic tree resulting from the SNP mining of the nuclear genome is similar to the chloroplast based phylogenetic tree with the same clades (Mussaendeae, Pavetteae, Bertiereae, Gardenieae, Coffeeae and Coffeeae/Bertiereae) being retrieved and highly supported even though the position of individual taxa within the clades may be different. Other relationships, such as the sister relationship between Ixoreae and Vanguerieae and between the Vanguerieae alliance and the Coffeeae alliance, are not retrieved in the nuclear phylogeny.

With the aim to use a dataset with less missing data, SNPs were filtered leading to a total of 1,726 sites (SNPs) retained for further analysis. The resulting tree (S3 Fig) shows a long branch for the Psilanthus-Coffea clade that may indicate a highly divergent evolution between these species and the rest of the ingroup. The tree further differs from the one based on 28.800 SNPs in that Bertiera is not sister to the Coffeeae but to the ingroup clade consisting of all species except for Coffeeae. Similarly to the 28,800 SNPs-based tree, branch support values are generally high. The clades Pavetteae, Mussaendeae, Gardenieae (excluding Schumanniophyton),

Bertiereae and Coffeeae are supported as monophyletic. However, while the main clades are similar, their relative position is not the same in the two analyses. Sherbournieae are not retrieved as monophyletic, indicating that the reduction of the number of SNPs should be done with care due to possible bias in markers genomic distribution.

## Conclusions

In this study we reported and analyzed the chloroplast genome sequences for 27 species of the Rubiaceae subfamily Ixoroideae using next-generation sequences (NGS. Plastid and nuclear genome phylogenies are well congruent with each other with an overall well-supported branch. Generally, the tribes form well-identified clades but the tribe Sherbournieae is shown to be polyphyletic. With continuously dropping prices and an increasing output and efficiency of bioinformatic tools, NGS appears to be now the best choice to study difficult or neglected plant families, tribes or genera. Our methodology used here combined plastid genome recon-struction and SNP mining of the nuclear genome and was successful for Ixoroideae. The same methodology should be extended to the two other Rubiaceae subfamilies (Cinchonoideae and Rubioideae). This would permit to clarify the relationships between Rubiaceae taxa and to bet-ter understand genome evolution in the family in relation to adaptive traits. The increased availability of more reference genomes other than *Coffea* genomes will facilitate and speed up this process.

## Supporting information

**S1 Table. Junction sequence divergence among 28 Rubiaceae.** Taxa are ordered alphabeti-cally within tribes. The genes considered at the border between the main regions LSC, IR and SSC are those identified in this study. The distances between genes and junctions are given in bp. IRA is not confirmed the assembly for three species (*Mussaenda pubescens*, *Feretia aerugi-nescens* and *Pavetta schumanniana*).
(XLSX)

**S1 Fig. Illumina read coverage in the IR region of *Mussaenda pubescens*.**
(TIFF)

**S2 Fig. Gene order and orientation visualized in some pairwise comparisons using Artemis Comparison Tool.**
(TIFF)

**S3 Fig. Maximum likelihood nuclear tree of 28 Ixoroideae based on 1,726 nuclear SNPs (RaxML with GTR model of substitution) and bootstrap values to estimate branch sup-ports.** Colored clades indicate well-defined tribes. The tree is rooted midpoint since no out-group outside Ixoroideae is available.
(TIFF)

**S1 Appendix. Fasta sequences of assembled cp genomes.**
(TXT)

## Acknowledgments

We thank the gardeners, Mr. H. Leqeux, Mr. J. Van Eeckhoudt, and the scientific curators, Ms. V. leyman and Dr. M. Reynders, for their care for the living Rubiaceae collection at Meise Botanic Garden. We thank Mr. F. Van Caekenberghe for help with the collection of plant material. We are grateful to the manager of the molecular laboratory at Meise Botanic Garden,

Mr. W. Baert, for his goodwill towards this project. We thank Meise Botanic Garden for funding sequencing work, travel and publication fees. We thank the Institut de Recherche pour le Développement (IRD) for funding sequencing work and for providing a fellowship for S.N. Ly. The Laboratory Chrono-Environment, Université de Bourgogne Franche-Comté, also funded part of the sequencing.

## Author Contributions

**Conceptualization:** Petra De Block, Perla Hamon, Romain Guyot.

**Data curation:** Serigne Ndiawar Ly, Andrea Garavito.

**Formal analysis:** Serigne Ndiawar Ly, Christophe Guyeux.

**Funding acquisition:** Petra De Block, Arnaud Mouly, Perla Hamon.

**Investigation:** Christophe Guyeux, Jean-Claude Charr.

**Methodology:** Pieter Asselman, Romain Guyot.

**Project administration:** Petra De Block, Perla Hamon, Romain Guyot.

**Resources:** Jean-Claude Charr, Steven Janssens, Romain Guyot.

**Software:** Christophe Guyeux, Jean-Claude Charr.

**Supervision:** Andrea Garavito, Petra De Block, Romain Guyot.

**Validation:** Petra De Block, Perla Hamon, Romain Guyot.

**Visualization:** Andrea Garavito, Romain Guyot.

**Writing – original draft:** Petra De Block, Perla Hamon.

**Writing – review & editing:** Serigne Ndiawar Ly, Andrea Garavito, Petra De Block, Pieter Asselman, Christophe Guyeux, Jean-Claude Charr, Steven Janssens, Arnaud Mouly, Perla Hamon, Romain Guyot.

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
