## [Decision Letter · Decision Letter 0]

18 Mar 2020

PONE-D-19-35274

Chloroplast genomes of Rubiaceae: comparative genomics and molecular phylogeny in subfamily Ixoroideae

PLOS ONE

Dear Dr. guyot,

Thank you for submitting your manuscript to PLOS ONE. After careful consideration, we feel that it has merit but does not fully meet PLOS ONE’s publication criteria as it currently stands. Therefore, we invite you to submit a revised version of the manuscript that addresses the points raised during the review process.

We would appreciate receiving your revised manuscript by May 02 2020 11:59PM. To enhance the reproducibility of your results, we recommend that if applicable you deposit your laboratory protocols in protocols.io, where a protocol can be assigned its own identifier (DOI) such that it can be cited independently in the future. For instructions see: http://journals.plos.org/plosone/s/submission-guidelines#loc-laboratory-protocols

We look forward to receiving your revised manuscript.

Kind regards,

Shilin Chen

Academic Editor

PLOS ONE

Journal Requirements:

2. Please amend either the abstract on the online submission form (via Edit Submission) or the abstract in the manuscript so that they are identical.

Reviewers' comments:

Reviewer's Responses to Questions

**Comments to the Author**

1. Is the manuscript technically sound, and do the data support the conclusions?

Reviewer #1: Yes

Reviewer #2: Yes

2. Has the statistical analysis been performed appropriately and rigorously? 

Reviewer #1: Yes

Reviewer #2: N/A

3. Have the authors made all data underlying the findings in their manuscript fully available?

Reviewer #1: Yes

Reviewer #2: Yes

4. Is the manuscript presented in an intelligible fashion and written in standard English?

Reviewer #1: Yes

Reviewer #2: Yes

5. Review Comments to the Author

Reviewer #1: The research article is overall good, but many minor errors need to be removed prior to publishing. Tables including supplementary tables contains may space and other spelling, scientific nomenclature errors and the tables format. Therefore, my suggestion is to revise the manuscript thoroughly.

Abstract should be rewritten according to the PLos ONE.

Many spelling errors should be revised, here only give some examples:

Line 3 of introduction: ‘......13.600 species......’ should be ‘.......13,600 species.......’?

‘......such as Vangueria, Alibertia and Duroia L.f.’: scientific nomenclature should be consistent.

Reviewer #2: the authors provided whole cp genome sequences for 27 species of the Rubiaceae subfamily Ixoroideae which sequeced using high-throughput genome sequencing method, analyzed the plastid genome structure and the chloroplast genome evolution in the Rubiaceae through efficient methodology for de novo assembly of plastid genomes. this work is meaningful for the molecular phylogeny analysis in Rubiaceae subfamily Ixoroideae.

Some suggestion:

1) What is the role played by the second main objectives of this paper, where "to test the efficiency of mining SNPs in the nuclear genome of Ixoroideae based on the use of a coffee reference genome to produce well-supported nuclear trees"，support the chloroplast genome results ?

2) The content of the discussion part and the result part are repeated, so it is suggested that merged these two parts.

6. PLOS authors have the option to publish the peer review history of their article (what does this mean?). If published, this will include your full peer review and any attached files.

Reviewer #1: No

Reviewer #2: No

---

## [Author Response · Author response to Decision Letter 0]

25 Mar 2020

PONE-D-19-35274 Responses to Reviewers. 

Thanks to reviewers for their valuable suggestions. Please find below our answers.

Reviewer #1: “The research article is overall good, but many minor errors need to be removed prior to publishing. Tables including supplementary tables contains may space and other spelling, scientific nomenclature errors and the tables format. Therefore, my suggestion is to revise the manuscript thoroughly”.

Ans: The manuscript has been revised thoroughly. 

“Abstract should be rewritten according to the PLos ONE”.

Ans: Done: Describe the main objective(s) of the study -> OK ; Explain how the study was done, including any model organisms used, without methodological detail -> Ok ; Summarize the most important results and their significance -> Ok ; Not exceed 300 words -> OK ; Abstracts should not include Citations -> Ok

“Many spelling errors should be revised, here only give some examples: Line 3 of introduction: ‘......13.600 species......’ should be ‘.......13,600 species.......’? ‘......such as Vangueria, Alibertia and Duroia L.f.’: scientific nomenclature should be consistent.”

Ans: Done. The changes have been highlighter along the manuscript. 

Reviewer #2: “the authors provided whole cp genome sequences for 27 species of the Rubiaceae subfamily Ixoroideae which sequeced using high-throughput genome sequencing method, analyzed the plastid genome structure and the chloroplast genome evolution in the Rubiaceae through efficient methodology for de novo assembly of plastid genomes. this work is meaningful for the molecular phylogeny analysis in Rubiaceae subfamily Ixoroideae.”

Some suggestion:

“1) What is the role played by the second main objectives of this paper, where "to test the efficiency of mining SNPs in the nuclear genome of Ixoroideae based on the use of a coffee reference genome to produce well-supported nuclear trees"，support the chloroplast genome results?”

Ans: Mining SNP using the C. canephora genome has been tested (Hamon et al., 2017) and was successful. An average of 79% of the SNP determined in the Coffea genus are transferable to Ixoroideae, with variation ranging from 35% to 96%. A sentence has been introduced in the Abstract and a chapter called “Nuclear SNP mining and Efficiency of transferability of methods from Coffea to Ixoroideae” in Results and Discussion“ has been reorganized to explain better this part. 

“2) The content of the discussion part and the result part are repeated, so it is suggested that merged these two parts.”

Ans: Done. We reorganize the manuscript with a Results and Discussion chapter.

---

## [Decision Letter · Decision Letter 1]

13 Apr 2020

Chloroplast genomes of Rubiaceae: comparative genomics and molecular phylogeny in subfamily Ixoroideae

PONE-D-19-35274R1

Dear Dr. guyot,

We are pleased to inform you that your manuscript has been judged scientifically suitable for publication and will be formally accepted for publication once it complies with all outstanding technical requirements.

With kind regards,

Shilin Chen

Academic Editor

PLOS ONE

Additional Editor Comments (optional):

Reviewers' comments:

Reviewer's Responses to Questions

**Comments to the Author**

1. If the authors have adequately addressed your comments raised in a previous round of review and you feel that this manuscript is now acceptable for publication, you may indicate that here to bypass the “Comments to the Author” section, enter your conflict of interest statement in the “Confidential to Editor” section, and submit your "Accept" recommendation.

Reviewer #1: All comments have been addressed

Reviewer #2: All comments have been addressed

2. Is the manuscript technically sound, and do the data support the conclusions?

Reviewer #1: Yes

Reviewer #2: Yes

3. Has the statistical analysis been performed appropriately and rigorously? 

Reviewer #1: Yes

Reviewer #2: Yes

4. Have the authors made all data underlying the findings in their manuscript fully available?

Reviewer #1: Yes

Reviewer #2: (No Response)

5. Is the manuscript presented in an intelligible fashion and written in standard English?

Reviewer #1: Yes

Reviewer #2: Yes

6. Review Comments to the Author

Reviewer #1: (No Response)

Reviewer #2: (No Response)

7. PLOS authors have the option to publish the peer review history of their article (what does this mean?). If published, this will include your full peer review and any attached files.

Reviewer #1: No

Reviewer #2: No

---

## [Editor Report · Acceptance letter]

17 Apr 2020

PONE-D-19-35274R1 

Chloroplast genomes of Rubiaceae: comparative genomics and molecular phylogeny in subfamily Ixoroideae 

Dear Dr. guyot:

I am pleased to inform you that your manuscript has been deemed suitable for publication in PLOS ONE. Congratulations! Your manuscript is now with our production department. 

With kind regards,

on behalf of

Dr. Shilin Chen 

Academic Editor

PLOS ONE